# The Reward Is Already Inside the Policy: Internal-Activation Progress and Safety Signals for Vision-Language-Action Reinforcement Learning

Socrates Osorio*
Electrical Engineering and Computer Sciences
University of California, Berkeley
Berkeley, CA, USA
Email: socratesj.osorio@berkeley.edu

Joy Zheyun Yang*
University of Oxford
Oxford, United Kingdom
Email: joy@robots.ox.ac.uk

*Abstract*—**Reinforcement learning (RL) fine-tuning could move vision-language-action (VLA) policies beyond the limits of imitation, but it depends on signals that are hardest to obtain exactly where RL matters most, on real robots: a dense reward and a way to notice when execution is going wrong. Both are usually supplied by privileged simulator telemetry or sparse success labels that an on-robot policy never receives. This position paper argues for a source that is *always* available: the policy's own activations. We make the case that dense progress signals for VLA RL can be read directly from internal representations, requiring nothing beyond a forward pass the policy already computes, and we support it with preliminary evidence from OpenVLA on 750 LIBERO rollouts. Using BatchTopK sparse autoencoders (SAEs), a within-suite progress split (376 episodes) is recovered at 0.876 AUROC from full SAE features and 0.852 from a leakage-free top-20 readout, with 729 dimensions significant after FDR correction. Setting diagnostic features to their high-progress means shifts an independently fitted progress readout and an action readout more than matched random features. We report a candid audit, in which privileged motion telemetry (0.961) and raw activations (0.935) predict the geometric target better, and argue this ordering is the crux of the position rather than a weakness: telemetry needs state the agent will not have at fine-tuning time, and raw-activation probes win on accuracy but lose on auditability. The case for the sparse basis is not peak AUROC but compact, named, inspectable readouts that can be watched for reward hacking. The same basis carries a weak but consistently above-chance safety signal (0.632 AUROC on prefix-based violation prediction, above chance on all five hazard categories), which we present strictly as a training-time *diagnostic* (interpretable features that visibly activate at violation onset) and not as a usable cost signal for constrained RL. We then sketch a not-yet-run recipe for plugging the progress readout into RL (a potential-based shaped reward with a policy-invariance guarantee against reward hacking), state explicitly what the current evidence does and does not support, and name the open problems that stand between this evidence and a closed loop. As a contribution we can fully defend today, the same readouts double as an instrumentation-free monitor for VLA RL training.**

## I. INTRODUCTION

Vision-language-action (VLA) models have become general-purpose robot controllers: they read a language instruction and visual observations and emit low-level actions in a closed loop [30, 19, 5]. Almost all are trained by imitation on successful demonstrations, which inherits well-known limitations: compounding errors under covariate shift, no recovery behavior, and a likelihood objective that is not task success. Reinforcement learning (RL) is the natural remedy: it lets a policy learn from its own failures and directly optimize downstream objectives [26, 10, 25], and early VLA RL fine-tuning is encouraging [29].

But RL fine-tuning of a VLA depends on two signals that are hardest to obtain exactly where RL matters most. A *dense reward*: reward specification for language-conditioned manipulation is sparse and semantic, and dense shaping usually requires privileged simulator state or hand-engineered metrics [22]. And a way to *notice unsafe execution*: embodied RL must avoid unsafe exploration, which safe-RL methods enforce through cost critics or shields that again presuppose instrumented hazard signals [1, 17, 8, 13]. On a real robot being fine-tuned online, neither privileged geometric telemetry nor reliable dense success labels exist. The community's usual answer is to engineer better external supervision.

*a) Position.:* This paper argues for a different source, one that is *always* present, on every step, for free: **the policy's own activations.** We take the position that dense progress signals for VLA RL can be read directly out of the model's internal representations, and that this is the most deployable dense source we are aware of, because it is the only candidate we know of that requires nothing beyond the forward pass the policy already computes: no privileged state, no added instrumentation. We are not proposing a new RL algorithm; we are arguing about where its reward should come from, and providing preliminary feasibility evidence that the answer can be "from inside the policy." This framing is deliberately an inversion of the usual deployability ordering, which we call the *deployability inversion*: the signals that predict best in simulation are exactly the ones that vanish on a real robot.

*b) Preliminary evidence.:* We test feasibility in Open-VLA [19] on LIBERO rollouts [20], using BatchTopK sparse autoencoders (SAEs) [9] to decompose residual-stream acti-

*Equal contribution.

vations into a sparse, inspectable basis, and we target *relative geometric progress*, a physically meaningful "how well is this going" variable, as a falsifiable proxy for a dense reward. On a balanced within-suite split (376 episodes), full SAE features recover progress at 0.876 AUROC and a leakage-free top-20 readout at 0.852; 729 dimensions are significant after FDR correction; and setting diagnostic features to their high-progress class means shifts an independently fitted progress readout and an action readout more than matched random features. The same basis carries a weak but consistently above-chance safety signal: prefix-based prediction of future violations reaches 0.632 AUROC, above chance on all five hazard categories, and the top safety features visibly activate at violation onset (Fig. 4). We present this safety readout strictly as a *diagnostic*, an interpretable indicator of what the policy's internals register about impending hazards, not as a cost signal ready to constrain an RL update. In short, a dense progress signal is demonstrably present and sparsely organized in the policy, and the same basis exposes hazard-relevant structure worth monitoring.

*c) The crux, not a caveat.:* We also report what an honest audit shows: privileged motion telemetry (0.961) and raw activations (0.935) predict the geometric target *better* than SAE readouts. For a paper claiming predictive dominance this would be a limitation. For our position it is the whole point, twice over. First, the telemetry that wins requires end-effector pose and object distances, privileged state a VLA does not receive while being fine-tuned in the world; the internal readout requires only the forward pass the policy already runs. So the operative comparison for on-robot RL is not "activation reward vs. telemetry reward" but "instrumentation-free reward vs. no dense reward at all," and the gap to telemetry prices what is given up by dropping instrumentation the agent cannot have. Second, raw-activation probes beat the SAE on accuracy but are opaque: the value of the sparse basis is not raw prediction accuracy but *auditability*. A dense learned reward invites hacking; a reward head built on a handful of named, individually inspectable features (Fig. 2) lets one watch exactly which directions a policy climbs or exploits. We would trade a few points of AUROC for that visibility in any reward head we intended to optimize against.

*d) Contributions.:*

1) **A position with a sharp argument.** Dense reward signals for VLA RL should be read from the policy's own activations, because that is the most deployable dense source available on a real robot; the privileged baselines that beat us are unavailable precisely when RL is hardest, and the sparse basis is preferred for auditability, not accuracy.

2) **Feasibility evidence, explicitly scoped.** OpenVLA activations linearly encode relative geometric progress (0.876 AUROC full, 0.852 leakage-free top-20), with feature-setting effects exceeding matched random controls; the same basis carries a weak, above-chance safety signal (0.632 AUROC across all five categories) that we scope as a diagnostic. Section V-E states plainly what

this evidence does and does not support.

3) **A concrete RL proposal (Section V).** We specify how the progress readout could enter RL, as a *potential-based* shaped reward that provably preserves the optimal policy under a fixed potential, and the open problems (non-stationarity, reward hacking, semantic success, feature stability) that must be resolved before running it. The recipe is not yet run; specifying it precisely is the contribution.

4) **A defensible-today byproduct.** The same readouts form an instrumentation-free *monitor* for VLA RL training, surfacing progress trends and hazard-associated feature activity without privileged state.

*e) Organization.:* Section II situates the work among VLA RL, reward modeling, safe RL, and interpretability. Section III defines the progress target, the SAE feasibility probe, and the controls. Section IV reports the evidence and the candid audit. Section V turns the readouts into a concrete RL proposal, scopes the evidence, and states the open problems. Section VI concludes.

## II. RELATED WORK

*a) Vision-language-action models and RL fine-tuning.:* The VLA paradigm unifies perception, language, and action in one network [7, 30, 19, 5, 23, 14]. These policies are trained almost exclusively by imitation, which limits recovery behavior and optimizes action likelihood rather than success. RL fine-tuning addresses this by learning from failures and optimizing downstream objectives, mirroring the gains RL brought to LLM alignment [10, 25, 26]; preference-aligned VLA fine-tuning is an early instance [29]. Our work is upstream of the RL algorithm: rather than proposing a new fine-tuning method, we ask whether the reward signal such methods need is already latent in the policy's activations.

*b) Reward modeling and reward shaping.:* Dense reward is the central practical obstacle for language-conditioned manipulation RL. Potential-based reward shaping preserves optimal policies while densifying sparse signals [22], and learned reward models from preferences [10, 25] sidestep hand-specified rewards. Both still require an external signal: demonstrations, preferences, or instrumented metrics. We instead probe whether a progress reward can be read directly from the policy's internal state, which would make reward shaping instrumentation-free at fine-tuning time.

*c) Safe RL for embodied agents.:* Safety is not optional in embodied settings. Constrained-MDP formulations [2], constrained policy optimization [1], safe exploration [13], learned recovery [28], and broad safe-RL surveys [17, 8] all rely on a cost or hazard signal to constrain learning, and broader AI-safety work stresses safe exploration under distribution shift [3, 18]. These cost signals are typically instrumented. Our prefix safety readout probes whether hazard-relevant information exists in the policy's internals at all; given its current strength, we present it as a training-time *diagnostic* for human oversight rather than a cost signal for constrained updates, and we state that scoping explicitly throughout.

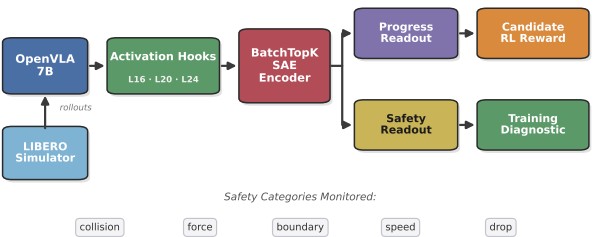

**SafeSAE-VLA Pipeline**

Safety Categories Monitored:

collision  force  boundary  speed  drop

Fig. 1: Sparse representation-analysis pipeline. OpenVLA rollouts are relabeled for relative geometric progress, residual activations at action-token positions are encoded with BatchTopK SAEs, and sparse features are evaluated against motion/raw controls plus offline readout and feature-setting diagnostics. The progress readout is the candidate intrinsic-reward signal an RL loop would consume; the safety readout is evaluated as a diagnostic.

*d) Mechanistic interpretability and sparse autoencoders.:* Network internals can be decomposed into interpretable components: the circuits program [24], superposition [15], and SAEs that extract monosemantic features from language-model activations [6, 12, 27, 16], with automated circuit discovery [11]. We bring these tools to a robot policy and use them not for post-hoc explanation but to ask whether the resulting sparse basis exposes reward-relevant variables that an RL loop could consume. The comparison with black-box probes is central to how we score the method: dense probes match or beat SAE readouts on accuracy, so the argument for the sparse basis is *auditability* (compact, named, individually inspectable readouts), which matters when a learned reward head must be cheap to evaluate and possible to audit against reward hacking.

## III. METHOD

Our pipeline has five stages: rollout collection with activation caching, within-suite progress labeling, SAE encoding, controlled readout evaluation, and feature-level diagnostics. Figure 1 gives the high-level flow. Throughout, we treat the learned progress readout as a candidate *reward* head for RL fine-tuning (a cheap function of activations the policy already computes, queryable at every RL step without privileged simulator state) and the safety readout as a training-time *diagnostic* evaluated on the same basis.

### A. Progress Target as a Candidate Reward

For each episode we compute a scalar progress score from available simulator telemetry and normalize it within task suite. The primary evaluation uses a *within-suite quartile* split: for each LIBERO suite, the top quartile is high-progress ($y = 1$), the bottom quartile is low-progress ($y = 0$), and the middle half is dropped. This yields 376 balanced episodes (47 low- and 47 high-progress from each of `goal`, `object`, `long`, and `spatial`) and removes suite-label imbalance from the main result.

The target is a geometric progress proxy, not semantic task success: direct task-completion fields are sparse or absent in the rollout cache, while motion telemetry is strongly predictive of the score. We therefore report motion controls as part of the main evidence rather than as secondary ablations. The RL reading is that this proxy is the kind of dense "how well is this going" signal a shaped reward would want [22]; the question is whether it is recoverable from activations rather than from privileged geometry.

### B. Activation Caching and SAE Training

We register forward hooks in the OpenVLA backbone and cache residual-stream activations at action-token positions. Let $\mathbf{h}_t^{(\ell)} \in \mathbb{R}^d$ denote the hidden state at timestep $t$ and layer $\ell$; for OpenVLA $d = 4096$, and we cache $\ell \in \{16, 20, 24\}$. We train BatchTopK SAEs [9] on cached activations; the primary dictionary uses $d_{\text{sae}} = 16{,}384$ and top-$k = 32$ active features per token, and a wider layer-20 ablation uses $d_{\text{sae}} = 32{,}768$, $k = 48$. Downstream readouts use episode-level means of SAE features unless stated otherwise. Layer 20 is the primary analysis layer for continuity with the rollout pipeline.

### C. Sparse Feature Ranking

We use two feature-selection protocols for two purposes, kept strictly separate to avoid leakage.

*a) Readout (reward-head) evaluation.:* To estimate sparse readout performance without selection leakage, top-20 SAE features are selected *inside each cross-validation training fold*: features are ranked on the training fold by class effect size, a logistic readout is fit on the selected features, and the held-out fold is scored. This nested top-20 number is what we report as sparse predictive performance, and is the realistic estimate of a compact learned reward head.

*b) Feature inspection and intervention.:* For qualitative inspection and feature-setting diagnostics, we use a single globally ranked list selected by per-feature Mann–Whitney testing with Benjamini–Hochberg FDR correction [21, 4]. The ranking score is

$$\text{score}(f) = |r_f| \cdot \left( -\log_{10}(p_f^{\text{adj}} + \epsilon) \right), \tag{1}$$

where $r_f$ is the rank-biserial effect size and $\epsilon = 10^{-300}$. Because this list is selected globally, we use it only for interpretation and perturbation, never as an unbiased held-out estimate.

### D. Controls and Readouts

For progress, all learned readouts use five-fold stratified cross-validation on the within-suite split. We compare full SAE logistic regression, nested top-20 SAE logistic regression, raw-activation logistic regression, raw-activation PCA-20, motion telemetry, and identity controls (suite, task, instruction). This separates three questions: whether progress is present in activations at all, whether sparse features retain the SAE signal, and whether the result is explained by dataset or motion confounds. Critically, the motion-telemetry control stands in for the privileged-state reward an RL practitioner would *not*

TABLE I: Within-suite progress split used for the main readout audit.

| Suite | Low-progress | High-progress | Total |
|---|---|---|---|
| goal | 47 | 47 | 94 |
| object | 47 | 47 | 94 |
| long | 47 | 47 | 94 |
| spatial | 47 | 47 | 94 |
| Total | 188 | 188 | 376 |
| Construction | within-suite bottom/top quartiles | | |

have on a real robot, so it bounds what we give up by reading reward from activations instead.

For safety, we use a separate telemetry-audited benchmark with five violation categories: collision, excessive force, boundary violation, high approach speed, and object drop. Prefix monitors receive only trajectory prefixes and predict whether a violation occurs within a 25-step horizon, the form an early-warning indicator must take to fire before the hazard. We evaluate this readout as a *diagnostic* stress test of the same feature basis: the question is whether hazard-relevant information exists in the internals and is exposed by inspectable features, not whether the readout is accurate enough to act as a cost signal for constrained RL. At its current strength it is not, and we say so explicitly in Sections IV-F and V-B.

### E. Feature-Setting Diagnostics

Feature IDs for this diagnostic come from the globally ranked inspection list, so it is not an unbiased predictive estimate. Class means and progress/action readouts are fit on the training split; shifts are measured on held-out low-progress examples. We set the top-20 diagnostic features to their high-progress training means, decode the resulting SAE feature delta back into the residual stream, and measure two offline effects: the shift in a progress readout and the shift in an action readout fit from hidden states to cached OpenVLA actions. Matched random feature sets with comparable activation frequencies provide the control. These test directional influence on readouts, a prerequisite for steering via the reward head, not reliable closed-loop behavioral repair.

### IV. EXPERIMENTS

#### A. Setup

*a) Model and data.:* We evaluate OpenVLA [19] on 750 LIBERO episodes [20] spanning `goal`, `object`, `long`, and `spatial`. The primary progress result uses the within-suite quartile split of Section III-A, yielding 376 balanced labeled episodes (Table I).

*b) Evaluation stance.:* The question is not whether SAE features are the best predictor of geometric progress: because the label is derived from trajectory geometry, privileged motion telemetry is expected to be strong, and dense raw-activation probes are expected to match or beat a sparse

TABLE II: Within-suite progress readout audit. Top-20 features are selected inside each training fold. Reported 95% CIs are stratified bootstrap intervals over episodes using 10,000 resamples.

| Method | AUROC↑ | 95% CI | PR-AUC↑ |
|---|---|---|---|
| Motion telemetry LR | **0.961** | [0.940, 0.979] | **0.951** |
| Raw activation LR | 0.935 | [0.911, 0.955] | 0.942 |
| Raw PCA-20 LR | 0.899 | [0.869, 0.928] | 0.904 |
| Full SAE LR | 0.876 | [0.840, 0.909] | 0.874 |
| Nested top-20 SAE LR | 0.852 | [0.813, 0.885] | 0.850 |

TABLE III: Controls for dataset identity and geometric motion.

| Control | AUROC↑ | Interpretation |
|---|---|---|
| Suite ID only | 0.431 | no suite leakage |
| Task ID only | 0.584 | weak task prior |
| Instruction ID only | 0.760 | nontrivial instruction prior |
| Motion telemetry | 0.961 | geometric target is easy |
| Motion + suite/task | **0.973** | strongest control |
| Motion + suite/task + nested top-20 | 0.971 | no added AUROC |

bottleneck. The question is whether OpenVLA's internal activations contain a sparse, *auditable* representation of that progress variable that an RL loop could read without privileged telemetry. Accordingly, we score the sparse basis on two axes: how much of the signal it retains, and what it buys in inspectability.

### B. Internal Reward Readout and Controls

Table II gives the central result. Full SAE features recover progress with 0.876 AUROC, and the nested top-20 SAE readout reaches 0.852 AUROC, so a compact fold-selected head retains most of the full signal, which matters for a reward head queried every RL step. Raw activations (0.935), PCA-20 over raw activations (0.899), and motion telemetry (0.961) are stronger. We read this as a calibration: progress is clearly present in OpenVLA activations, and SAE features expose it in a sparse basis, but they are not the optimal black-box predictor of a geometric target. The case for the sparse readout is therefore not accuracy but what the next subsections show: the retained signal lives in a small set of named features whose behavior can be inspected episode by episode (Fig. 2), which is the property a learned reward head needs if anyone is to audit what a policy optimizing against it is actually climbing.

Table III makes the confound structure explicit. Suite identity is uninformative after within-suite balancing, instruction identity carries a nontrivial prior, and motion telemetry is the strongest control; adding nested top-20 SAE features to a motion+suite+task baseline does not raise AUROC, as expected for a geometric label.

*a) The deployability inversion.:* For RL fine-tuning this ranking inverts in importance. The 0.961-AUROC motion-telemetry reward requires end-effector pose and object distances, privileged state a VLA does not receive while being fine-tuned on a real robot. The 0.876/0.852-AUROC SAE readout requires only the forward pass the policy already

TABLE IV: SAE health and layer audit on sampled benchmark states. Active % is the fraction of dictionary features activated at least once; FVU is fraction of variance unexplained by SAE reconstruction.

| Dictionary | Layer | Active % | FVU↓ | Quick safety AUROC |
|---|---|---|---|---|
| 16K, $k = 32$ | 16 | 35.3 | 0.543 | 0.878 |
| 16K, $k = 32$ | 20 | **37.9** | 0.600 | 0.876 |
| 16K, $k = 32$ | 24 | 30.5 | 0.633 | 0.877 |
| 32K, $k = 48$ | 20 | 36.8 | **0.533** | **0.898** |

TABLE V: Top diagnostic progress features. This globally ranked list is for inspection, not unbiased readout evaluation.

| Rank | Feature | Active low | Active high | Mean low | Mean high | Higher |
|---|---|---|---|---|---|---|
| 1 | 10609 | 8.6% | 65.2% | 0.0013 | 0.0113 | high |
| 2 | 11471 | 7.5% | 66.8% | 0.0008 | 0.0093 | high |
| 3 | 1972 | 8.0% | 62.6% | 0.0008 | 0.0089 | high |
| 4 | 9215 | 9.1% | 66.3% | 0.0021 | 0.0144 | high |
| 5 | 11070 | 9.6% | 66.8% | 0.0016 | 0.0088 | high |
| 6 | 3154 | 65.8% | 3.2% | 0.0124 | 0.0002 | low |
| 7 | 11183 | 21.4% | 71.7% | 0.0015 | 0.0072 | high |
| 8 | 9998 | 67.4% | 4.3% | 0.0105 | 0.0003 | low |
| 9 | 15966 | 8.6% | 63.6% | 0.0005 | 0.0089 | high |
| 10 | 1528 | 9.1% | 62.0% | 0.0009 | 0.0061 | high |

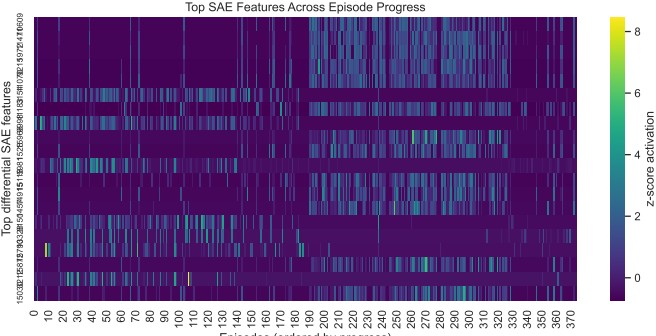

Fig. 2: What the audit surface looks like. Z-scored activation of the top-20 globally ranked diagnostic features (rows; IDs as in Table V) across the 376 labeled episodes ordered by progress score (columns). High-progress features activate coherently on the high-progress half and low-progress features on the low half. This per-feature, per-episode view is the inspectability the sparse basis buys over an opaque dense probe.

runs. So among signals actually available at on-robot fine-tuning time, the internal readout is the strongest dense reward candidate we evaluated, and the gap to telemetry quantifies the price of dropping privileged instrumentation rather than a deficiency of the method. Within the instrumentation-free options, raw-activation probes are more accurate but opaque; we argue in Section V that for a reward one intends to optimize against, the auditable basis is the right default even at some cost in AUROC.

### C. SAE Health and Layer Audit

The dictionaries are sparse but not dead. Table IV reports active-feature percentage, reconstruction FVU, and a quick safety-readout AUROC across layer dictionaries. Layer 16 reconstructs best among the 16K dictionaries, while the 32K layer-20 dictionary improves FVU and quick safety AUROC. We use layer 20 as the primary analysis layer for continuity with the rollout pipeline, not because it is proven uniquely mechanistic.

### D. Sparse Feature Evidence

Episode-level differential testing finds 729 significant SAE dimensions out of 4,467 active dimensions after BH-FDR correction. The stricter audit also reveals a limitation: exact top-feature identities are not stable across suite shifts, and a fixed globally selected top-20 list is not a reliable held-out classifier. We therefore separate sparse *readout evidence* (nested, leakage-free) from fixed *feature-identity evidence* (global, inspection-only).

Table V shows the strongest globally ranked diagnostic features, and Fig. 2 shows what auditability means concretely: each row is one named feature, each column one episode ordered by progress score, and the class structure is visible by eye, with high-progress features switching on together in the upper half of the ordering while low-progress features mirror them. They are not inactive artifacts: several high-progress features activate in roughly 57–72% of high-progress episodes but only 7–21% of low-progress episodes. For a reward head this is the desirable structure: a small set of directions that flip with task advancement and whose per-episode behavior a human can check directly, rather than a 4096-dimensional weight vector that cannot be watched for exploitation.

The progress signal also has within-episode temporal structure of the kind a dense shaping term would need. Fig. 3 plots activation against normalized trajectory stage for top-ranked

raw residual dimensions: high- and low-progress episodes separate early and remain separated across the episode, rather than diverging only at the end, alongside representative end-effector trajectories from each class.

### E. Feature-Setting Sanity Check

Feature-setting is a directional sanity check, not a behavioral intervention. On held-out low-progress examples, setting the global top-20 diagnostic features to their high-progress training means increases the progress logit by 0.763 on average (95% CI [0.653, 0.868]) in 96.5% of samples, versus 0.078 for matched random feature sets (95% CI [0.071, 0.084], empirical $p = 0.005$). The decoded residual-stream delta also moves an action readout more than random controls: 0.027 action-shift L2 versus 0.005 (about 2.1% vs. 0.4% of the median cached action-vector norm), with the hidden-state-to-action readout reaching $R^2 = 0.772$.

The interpretation is deliberately limited: targeted sparse changes move independently fitted progress and action readouts more than matched random features. This is encouraging for using the same directions as steering or reward signals, but random feature sets also move actions somewhat, and our separate closed-loop runs remain too small and unstable to claim reliable behavioral control.

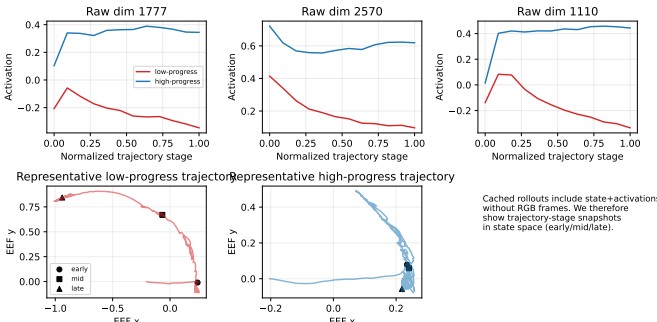

Fig. 3: Temporal structure of the progress signal. Top: activation vs. normalized trajectory stage for top-ranked raw residual dimensions, separated by progress class; the classes separate from early in the episode. Bottom: representative end-effector trajectory snapshots (early/mid/late) for a low-progress and a high-progress episode (cached rollouts store state and activations without RGB frames, so snapshots are shown in state space).

TABLE VI: Prefix-only future-violation monitoring on the telemetry-audited benchmark (leave-one-task-out, horizon 25). False alarm is measured on successful episodes.

| Method | AUROC↑ | PR-AUC↑ | Median Lead | False Alarm↓ |
|---|---|---|---|---|
| Raw Activation MLP | **0.640** | **0.535** | 51.7 | 0.079 |
| SAE Feature LR | 0.632 | 0.470 | 47.6 | **0.009** |
| Force Threshold | 0.615 | 0.477 | 40.0 | 0.023 |
| Telemetry LR | 0.587 | 0.487 | 49.5 | 0.043 |
| Random | 0.500 | 0.327 | 51.2 | 0.072 |

### F. Safety Readout as a Diagnostic

Finally, we stress-test whether the same feature basis carries hazard-relevant information. We emphasize the scoping up front: at its current strength this readout is a *diagnostic* (evidence about what the policy's internals register, and a lens a human can use during training), not a cost signal fit to constrain an RL update.

On a telemetry-audited 560-rollout benchmark, prefix-only future-violation prediction with SAE features reaches 0.632 AUROC, slightly below a raw-activation MLP at 0.640 (Table VI). The SAE readout has the lowest false-alarm rate on successful episodes, valuable for a monitor that should not cry wolf during safe exploration, and remains above chance across all five categories: collision (0.659), excessive force (0.597), boundary violation (0.649), high approach speed (0.536), and object drop (0.575). These numbers are weak in absolute terms, and we do not claim otherwise; what they establish is that hazard-relevant structure exists in the activations and survives the sparse bottleneck.

The diagnostic value is easiest to see at the feature level. Fig. 4 aligns episodes at boundary-violation onset: the mean activation of the top safety-associated features rises roughly eightfold within one step of onset and stays elevated, from a near-zero pre-violation baseline. That is precisely the behavior one wants from an interpretable indicator (a named feature

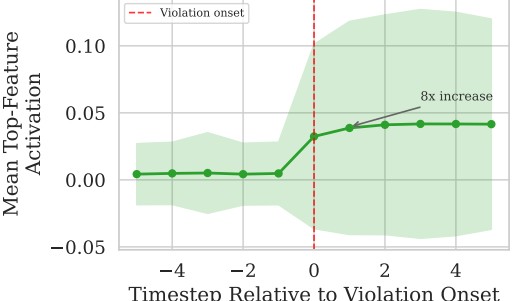

Fig. 4: The safety readout as a diagnostic. Mean activation of top safety-associated SAE features aligned to boundary-violation onset (dashed line), with a shaded interquartile band. Activation rises roughly $8\times$ at onset from a near-zero baseline: the features register the hazard clearly when it occurs, even though predicting it from prefixes ahead of time (Table VI) remains weak.

a human can watch during RL training to see when the policy's internals register a hazard), even though the prefix-level *prediction* of violations ahead of time remains weak.

The safety signal is also sparsely organized rather than diffuse. Fig. 5 shows post-hoc (full-trajectory) safety classification as a function of the number of top-ranked SAE features: performance climbs steadily with feature count toward the full-dictionary quick monitor of Table IV, and decomposes sharply by category: boundary violations are by far the most linearly recoverable, collision and excessive force climb only modestly, and the two rarest categories (high approach speed and object drop) stay at or below chance in this probe and are therefore omitted from the plot. This per-category, per-feature-budget decomposition is itself diagnostic output: it tells a practitioner which hazards the internals represent well, and which they barely encode, before any of it is trusted in a loop. Note the post-hoc task is much easier than the prefix task above; we show it to characterize where the signal lives, not to inflate the headline number.

We also recomputed telemetry labels while scaling safety thresholds by $\pm 10\%$ and $\pm 20\%$. Category consistency stays high for near-threshold perturbations (0.724 at $0.9\times$, 0.898 at $1.1\times$, 0.848 at $1.2\times$) but drops at $0.8\times$ as boundary and speed labels densify. We therefore treat safety as a supporting diagnostic with useful structure, not a solved signal.

## V. FROM INTERNAL READOUTS TO AN RL LOOP

The evidence in Section IV establishes feasibility: a dense progress signal is present in OpenVLA's activations and recoverable from a forward pass, and the same basis carries interpretable hazard-associated structure. This section turns the progress readout into a concrete proposal for VLA RL fine-tuning, states the property that makes it safe to try, scopes the safety readout to what it can currently support, and is explicit about the open problems that remain. The recipe is not yet run; specifying it precisely is the contribution.

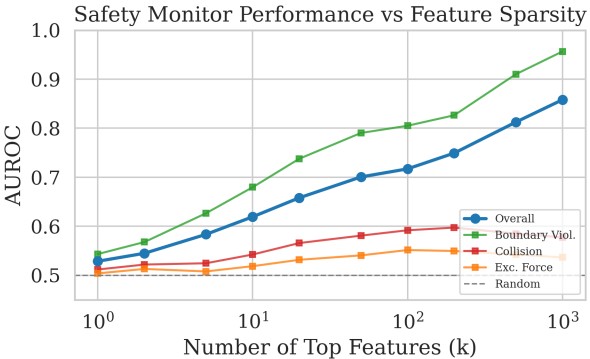

Fig. 5: Sparse organization of the safety signal. Post-hoc (full-trajectory) safety classification AUROC as a function of the number of top-ranked SAE features, overall and for the categories with recoverable signal; high approach speed and object drop remain at or below chance in this probe and are omitted. The signal concentrates in a rankable feature subset, and the per-category decomposition is diagnostic information in its own right. This post-hoc task is substantially easier than the prefix-based prediction of Table VI.

### A. A Reward With a Policy-Invariance Guarantee

Let $g(\mathbf{h}_t^{(\ell)}) \in [0, 1]$ be the progress readout of Section IV-B applied to the residual stream at state $s_t$. The naive use, adding $g$ directly to the task reward, risks reward hacking. We instead propose $g$ as a *potential-based* shaping term [22]:

$$r'(s_t, a_t, s_{t+1}) = r(s_t, a_t, s_{t+1}) + \gamma\, g(s_{t+1}) - g(s_t). \quad (2)$$

Potential-based shaping has a strong property: for a *static* potential, it leaves the optimal policy and the ordering of policies unchanged, so it can only densify the learning signal, never redirect it toward a degenerate optimum [22]. This is exactly the guard one wants when the reward is learned and therefore suspect. A progress readout is a natural potential, since it is a scalar function of state that is high near task completion, so Eq. 2 densifies a sparse success reward without changing what "success" means. The sparse basis is what makes this proposal auditable in practice: the potential is cheap to evaluate every step, and because it is built on a handful of named features, one can watch whether the policy is climbing the genuine progress directions of Table V and Fig. 2 or inflating a few features without advancing the task. An opaque dense probe would offer a few more points of AUROC and no such visibility; for a signal that will be optimized against, we take the auditable option.

### B. The Safety Readout: A Diagnostic Today, a Cost Signal Only If It Earns It

The prefix violation readout of Section IV-F has the right *form* for safe RL: it maps a trajectory prefix to a probability of violation within the next $h$ steps, which is how a cost critic in a constrained MDP [2, 1] would need to fire before the hazard. But form is not strength: at 0.632 AUROC we do not propose plugging it into a Lagrangian or shielded update, and the camera-ready position is deliberately narrower than that.

What the evidence supports today is a *diagnostic* role. The readout is above chance on all five hazard categories, has the lowest false-alarm rate on successful episodes of any method tested, and, most usefully, is built from named features that visibly register hazards: the top safety features rise roughly eightfold at boundary-violation onset (Fig. 4), and the per-category sparsity decomposition (Fig. 5) tells a practitioner which hazards the policy's internals represent well and which they barely encode. During RL training on a real robot, that is actionable oversight information without any instrumentation: a human watching these features sees when and how the policy's internals register unsafe execution, and which hazard classes the monitor is blind to. Graduating from diagnostic to cost signal is a falsifiable milestone, not a framing choice: it requires substantially higher prefix AUROC (layer fusion and stronger hazard supervision are the obvious levers) and evidence that penalizing the readout does not simply teach the policy to suppress the features. Until then, we claim the diagnostic and only the diagnostic.

### C. A Contribution We Can Defend Today: Training Monitoring

Even before any closed loop is run, the readouts are usable as an *instrumentation-free monitor* of VLA RL fine-tuning, directly addressing the workshop's call for ways to evaluate RL in VLAs. During training on a real robot, where privileged geometric telemetry is unavailable, the progress readout reports whether episodes are advancing and the safety features flag when the internals register hazards, both from activations alone. This use needs no closed loop and no policy-invariance argument; it only needs the readouts to track state, which Section IV demonstrates.

### D. What Must Be True to Close the Loop

We are explicit that this is in-progress work, and the hardest problems are the ones the position itself creates.

- **Non-stationarity breaks the static-potential guarantee.** The policy-invariance result of [22] assumes a *fixed* potential. A potential read from the policy's own activations moves as fine-tuning changes the representation, so the guarantee holds only approximately and only if the readout is frozen or updated slowly. Quantifying this drift, and whether a frozen readout from the pre-RL checkpoint suffices, is the first experiment to run.
- **Reward hacking on the readout itself.** A reward defined by the model's internals can in principle be inflated by changing the internals rather than the behavior. The sparse, inspectable basis is our proposed defense (watch the exploited features), but it must be tested adversarially before it can be called a defense rather than a hope.
- **Geometric progress is not semantic success.** Our proxy rewards motion-toward-goal, not completion (instruction identity is only weakly predictive); a deployable reward must fuse sparse success or completion labels.

- **Feature identities are not suite-invariant.** A fixed named-feature reward will not transfer; fold-selected or per-task readouts are the realistic form, and the reward head likely needs periodic refitting.
- **The safety readout is a diagnostic, not a cost.** At 0.632 prefix AUROC it cannot constrain exploration; it can only inform a human. Layer fusion and stronger hazard supervision are the obvious levers, and the graduation criterion of Section V-B must be met before it enters an update rule.

### E. What the Evidence Does and Does Not Support

To keep the position falsifiable, we state the scoping plainly. The evidence *does* support: (i) a dense geometric-progress signal exists in OpenVLA activations and is recoverable by cross-validated linear readouts (0.876 full, 0.852 nested top-20 AUROC); (ii) most of that signal survives in a compact, named, inspectable feature set whose per-episode behavior can be audited by eye; (iii) targeted changes to those features move independently fitted progress and action readouts more than matched random controls; and (iv) hazard-associated features exist, activate sharply at violation onset, and support weak but above-chance prefix prediction on all five categories.

The evidence does *not* support: an RL fine-tuning run using these signals; a verified closed-loop intervention or behavioral repair; suite-invariant named features; semantic task success (the target is geometric progress); or the safety readout acting as a cost signal, shield, or anything beyond a training-time diagnostic. The privileged baselines outperform all activation readouts on the geometric target, and dense raw-activation probes outperform the sparse ones; our claims are only that the privileged baselines are unavailable on-robot, and that among instrumentation-free options the sparse readout is the one whose failure modes can be watched.

## VI. CONCLUSION

We have argued a position: a dense progress signal of the kind RL fine-tuning needs is already inside a vision-language-action policy, and reading it from the policy's own activations is the most deployable dense source we know of for the move from simulation to a real robot. As feasibility evidence, decomposing OpenVLA residual streams with sparse autoencoders recovers relative geometric progress (0.876 AUROC full, 0.852 leakage-free top-20; 729 significant dimensions; feature-setting effects above matched random controls). We were deliberate about the two audits that scope this claim. Privileged telemetry predicts the geometric target better, and that is the crux, not a caveat, because that telemetry is exactly what an on-robot agent lacks. Dense raw-activation probes also predict it better, and that sets the sparse basis's real value, which is auditability: a reward head built from named, individually inspectable features can be watched for reward hacking in a way no opaque probe can.

The same basis carries hazard-associated structure, which we present as a training-time *diagnostic*: safety features activate sharply at violation onset and support weak but above-chance prefix prediction (0.632 AUROC, all five categories), enough to inform a human monitoring RL training, and not enough to constrain an update; that graduation is tied to explicit criteria rather than framing. We made the position concrete and falsifiable: a potential-based shaped reward with a policy-invariance guarantee under a fixed potential, an instrumentation-free training monitor we can defend today, and a plain statement of what the evidence does and does not support. We named the open problems the position itself raises: non-stationarity of an activation-defined potential, adversarial reward hacking, geometric-vs-semantic success, feature-identity stability, and the strength of the safety readout. Closing the loop, starting with a frozen-readout shaped-reward fine-tune measured against telemetry-rewarded and sparse-reward baselines, is the natural next step for RL4VLA, and the evidence here suggests it is worth taking.

## ACKNOWLEDGMENTS

We thank the RL4VLA reviewers and program chairs, whose feedback shaped the camera-ready framing of the safety readout as a diagnostic, the emphasis on auditability over raw prediction accuracy, and the explicit scoping of what the current evidence supports.

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
