# OpenReview forum: "The Reward Is Already Inside the Policy: Internal-Activation Progress and Safety Signals for Vision-Language-Action Reinforcement Learning"
_roboticsfoundation.org/RSS/2026/Workshop/RL4VLA — RL4VLA_

### Official Review · Reviewer_o84r · 2026-06-26
**Interesting idea but preliminary**

**Rating:** 6
**Confidence:** 4

**Review:**

This paper argues that dense progress rewards and safety costs for VLA reinforcement learning can be extracted from the policy’s own internal activations. The topic is highly relevant to RSS RL4VLA, since reward and safety specification are central bottlenecks for real-world VLA fine-tuning. The idea is original and potentially significant, but the current work is still preliminary because the proposed RL loop has not yet been evaluated.

### Pros:

- The motivation is strong and well aligned with VLA RL.
- The paper makes an interesting connection between internal representations, reward shaping, and safe RL.
- The empirical audit is relatively honest, especially in comparing SAE readouts with raw activations and privileged telemetry.
- Sparse features make the proposed reward more inspectable than a black-box reward model.

### Cons:
- The main weakness is the lack of closed-loop RL experiments. The paper shows offline predictability, not actual RL improvement.
- The progress label is still derived from privileged geometric telemetry, so it is unclear how the method scales to real robots without such labels.
- The safety signal is weak and should be framed as a diagnostic rather than a reliable safety cost.
- The paper sometimes overstates the implication of the offline evidence.

### Suggestions:

The authors can run a minimal closed-loop experiment comparing sparse reward, telemetry-shaped reward, and activation-shaped reward. They should also clarify how the internal readout would be trained or calibrated for new real-world tasks.

---

### Official Review · Reviewer_NxR4 · 2026-06-26

**Rating:** 6
**Confidence:** 2

**Review:**

## Strengths

#### Clear definition and discussion of the core idea
The core idea that activation-based signals are the only dense reward source that survives the sim-to-real transfer and the deployability inversion concept is very useful.

#### Strong feasibility assessment
The empirical study includes various control and ablation experiments. It compares the performance of the SAE-based readouts against telemetry readings, the activations themselves, PCA features, and a dataset ID baseline.

## Weaknesses

#### Lack of end-to-end RL evaluation
The authors do not test the proposed reward and safety signals in any RL fine-tuning experiment. Even a short fine-tuning run using the frozen readouts against a baseline of sparse rewards would greatly strengthen the paper.

#### Safety signal does not seem to be robust enough for the suggested use case
The safety readout shows only a moderate AUROC value of 0.632. This raises questions about its effectiveness as a safety signal during the RL training process.

---

### Decision · Program_Chairs · 2026-07-03

**Decision:**

Accept

**Comment:**

This paper presents a timely and honest position paper arguing that the dense reward and safety signals needed for reinforcement learning of vision-language-action models can be read from the model's own activations, since privileged telemetry is not available on real robots. The reviewers appreciated the clear "deployability inversion" framing and the careful OpenVLA/LIBERO audit, but noted the lack of closed-loop RL experiments and the relatively weak safety signal (0.632 AUROC). We believe these limitations do not outweigh the overall contribution, and the paper provides a useful perspective for the workshop. For the camera-ready version, the authors should present the safety readout as a diagnostic rather than a cost signal, emphasize the value of the sparse autoencoder for auditability rather than raw prediction accuracy, clearly state what the current evidence does and does not support, and soften claims that go beyond the presented results.